# Divide And Conquer: Efficiently Decoupling Consensus And Divergence For Federated Large Language Model Fine-Tuning

## Abstract

Federated Learning provides an efficient framework for fine-tuning Large Language Models (LLMs) on diverse private datasets, addressing the growing scarcity of publicly available training data while maintaining data privacy. However, in practice, client data typically spans multiple domains, posing significant challenges for the global model's generalization capabilities. To address this issue, we introduce a novel framework, **Fed**erated **C**onsensus-**D**ivergence **D**ecoupling for LLM Fine-Tuning (**FedCDD**), designed to enhance global model performance in such heterogeneous environments. Our framework introduces a mechanism for consensus aggregation and divergence alignment, decoupling client updates into "consensus" and "divergence" parts. This allows the LLM to maintain a unified consensus while accommodating domain-specific divergences. Additionally, we employ a Gaussian-Noise Mask to regulate local model uploads, preventing the LLM from overfitting to domain-specific knowledge. Experimental results on heterogeneous datasets demonstrate the superiority of our approach over existing methods. The code is anonymously available at https://anonymous.4open.science/r/FedCDD-5DA6.

## 1 Introduction

Trained on large public datasets, Large Language Models (LLMs) (Achiam et al., 2023) (Ouyang et al., 2022) have demonstrated significant success in solving general problems (Imani et al., 2023) (Didolkar et al., 2024), (Chen et al., 2023). However, the availability of high-quality public data is diminishing, posing a serious obstacle to the continued development of LLMs (Kaddour et al., 2023), and it is predicted that high-quality public data will be exhausted before 2026 (Villalobos et al., 2022). As a result, there is a growing trend of either combining existing datasets (Wang et al., 2023) or using datasets generated by models themselves (Wang et al., 2022). The former often falls short, as more data generally leads to better performance (Kaplan et al., 2020), while the latter may cause model degradation (Alemohammad et al., 2023) (Muennighoff et al., 2023). Meanwhile, many high-quality private datasets exist but cannot be shared due to privacy concerns. Some large language models, such as BloombergGPT (Wu et al., 2023), have been successfully trained on such datasets. The challenge lies in utilizing these private, high-quality datasets while preserving privacy. Federated Learning (FL) offers a solution by allowing multiple parties to collaboratively train a model without directly sharing their datasets. Participants train local models on private datasets, and only model updates are aggregated centrally, ensuring privacy (Li et al., 2020). Applying federated learning to Large Language Models offers a solution to the limitations of public high-quality datasets. It unlocks the potential value of private datasets without directly sharing them, thereby ensuring the privacy and security of the model data. (Zhuang et al., 2023).

With the increasing integration of Federated Learning (FL) and Large Language Models (LLMs), numerous studies have concentrated on training LLMs within a federated learning framework. In this paper, we focus on the integration of the LLM Supervised Fine-Tuning (SFT) module within the FL domain (Gunel et al., 2020). Existing approaches, such as FederatedScope-LLM (Kuang et al., 2024) and Shepherd (He et al., 2021), attempted to incorporate the FedAvg algorithm into the SFT process. However, subsequent research by Ye et al. (2024) pointed out that these studies were limited by using only one dataset and relying solely on FedAvg, thereby overlooking other scenarios.

Consequently, some research has explored the feasibility of the SFT module with a broader range of datasets and FL methods. Nevertheless, a common limitation in these studies is their failure to adequately address the diversity of real-world datasets, which can result in substantial client drift. While some studies have attempted to train clients on diverse datasets during the SFT process, they do not explicitly optimize for this scenario or propose methods tailored to this challenge, instead relying on conventional federated learning approaches. To the best of our knowledge, we are the first to study the integration of FL and SFT on Non-IID datasets and propose a innovative framework for LLM fine-tuning in FL. We aim to explore optimization strategies to enhance the performance of LLM SFT. By training on multiple datasets, we seek to closely simulate real-world environments, provide relevant benchmarks for this domain, and offer preliminary contributions that may inspire future research.

In the common experiment with federated large language model training on the Non-IID datasets, the proposed approach involves clients fine-tuning their models using Low-Rank Adaptation (LoRA) (Hu et al., 2021), then uploading the trained results to the cloud. The server aggregates models with FedAvg and distributes the updated global model back to the clients for the next round of training. In the server aggregation phase, we observe that the LoRA method can cause knowledge drift. The local LLM tends to focus on the local domain, which negatively impacts the global model's generalization ability. Based on this observation, we raise the following question: ***1) During the global aggregation, how to design a new method that ensures the global model accurately captures more details of local knowledge?*** In terms of client-side updates, we find that extracting local features during client training is crucial for the global model. Great training method should be better capture features from datasets while avoiding to upload the unimportant knowledge. However, in the current method, the best algorithm proved by Ye et al. (2024) simply uploads all the LoRA parameters to the global. Based on this, we further propose another question: ***2) During client training, how to employ a new algorithm that allows clients to provide the important knowledge extracted from local dataset for global?***

To address the two issues mentioned above, this paper proposes a innovative framework to help global model to absorb the knowledge from clients, enhancing generalization. To address problem 1, we explore the role of LoRA in federated learning, decomposing it into consensus characteristics and divergence characteristics. The former reflects the generalization ability of knowledge, while the latter indicates its divergence on specific domains. Based on this, we propose the Consensus-Divergence Aggregation, which combines consensus aggregation and divergence alignment, optimizing the global aggregation process and improving the performance of the global model. To the specific problem 2, during local updates, we introduce the Gaussian-Noise Masking, which focusing upload significantly altered parameters while disregarding minimal changes. The mask enables the global model to accurately capture the direction of meaningful updates, reducing the risk of entrapment in local optima caused by minor parameter adjustments.

Our primary contribution in this paper can be summarized as follows:

(1) We discover that LLM could be decoupled by consensus and divergence, which has practical significance in the federated large language model training.

(2) We propose an innovative framework for federated large language model training. In the global aggregation phase, we address the issue of domain knowledge drift in LLM clients by employing a combination of consensus aggregation and divergence alignment. During the client uploading process, we focus on ensuring that the parameter updates effectively contribute to the global model's knowledge base.

(3) Our framework applied in the same experimental environment significantly improves the model's generalization ability.

## 2 RELATED WORK

### 2.1 PARAMETER EFFICIENT FINE-TUNING IN LARGE LANGUAGE MODELS

Large Language Model (LLM) is a computational model capable of language generation or other natural language processing tasks. LLMs such as GPT-4 (Achiam et al., 2023), LLama3 (Dubey et al., 2024) (Touvron et al., 2023), Claude 3 has displayed great potential in various fields. Gener-

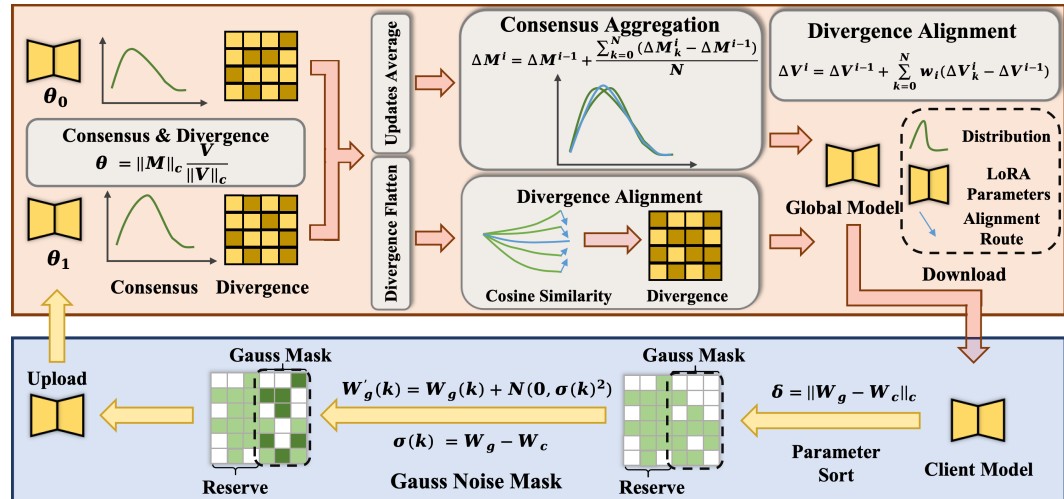

Figure 1: **Architecture illustration** of the Aggregation and Masking components. The two key components are shown at the top (a) and bottom (b) of the image, with nodes of different classes marked in different colors. (a) Consensus-Divergence Aggregation (Section 3.2.1) module splits the LoRA updates into consensus and divergence parts, applying consensus aggregation through delta averaging and divergence alignment using cosine similarity. (b) Gauss-Noise Masking (Section 3.2.2) component adds a mask in the client upload process, selecting important knowledge from LLM clients for updates. Best viewed in color. Zoom in for details.

ally, the process of training a LLM including: (1) Train the base model on the large dataset, such as Pile (Gao et al., 2020), LLaMA (Touvron et al., 2023) and etc. (2) Use Supervised Fine-Tuning (SFT) method (Xu et al., 2023a) (Brown, 2020) (Chen et al., 2024) to make LLM follow human's instruction. (3) Use Reinforcement Learning from Human Feedback (RLHF) (Sun et al., 2023) to align the model on the human-annotated or AI-annotated preference dataset, making LLM understand the human's value.

Fine-tuning is critical for adapting large language models (LLMs) to downstream tasks, but it requires significant computational resources (Houlsby et al., 2019) (Valipour et al., 2022) (Mao et al., 2021). As a result, Parameter Efficient Fine-Tuning (PEFT) methods (Xu et al., 2023b) are commonly used in LLM training to better fit downstream domains, particularly for speeding up processes such as SFT and RLHF. These methods require only small parameter updates compared to updating all the parameters of the pre-trained model, significantly reducing the computational overhead. Common implementations of PEFT include adapter techniques (Houlsby et al., 2019) (Pfeiffer et al., 2020) (He et al., 2021) (Edalati et al., 2022), prompting methods (Petrov et al., 2023) (Li & Liang, 2021), and Low-Rank Adaptation (LoRA) (Hu et al., 2021) (Valipour et al., 2022).

However, the adapter structure will introduce additional computational overhead (Houlsby et al., 2019) (Hu et al., 2023). The prompt method is unstable, difficult to fine-tune and cannot take long input sequences (Li & Liang, 2021). Nowadays, the LoRA structure is commonly used as the primary PEFT method in the SFT process, as it reduces computational costs without adding inference latency (Hu et al., 2021).

LoRA is a method that focuses on the low "intrinsic rank" of weight changes during model adaptation (Hu et al., 2021). LoRA freezes the base model and uses the format below to update the low "intrinsic rank." In this way, LoRA only requires updating a small set of parameters compared to the traditional approach, thereby accelerating the model $\theta$ training process.

$$\theta' = \theta + \Delta W = \theta + BA$$

In our paper, we focus on the SFT process, with the goal of exploring the practical significance of LoRA in the integration of FL and LLM training. This approach aims to optimize our experimental framework more effectively.

## 2.2 Federated Learning and Large Language Models

Federated Learning (FL) (Kairouz et al., 2019) is a method that holds great potential for enabling privacy-preserving collaborative training. FL involves four key processes during training: (1) The server sends the global model to the client side; (2) The client performs local model training on its local dataset; (3) The client uploads the updated model parameters; and (4) The server receives the client model parameters and performs global aggregation. Through these iterative steps, the server is able to train a global model without requiring clients to directly share their local datasets.

In response to the shortage of high-quality public datasets, FL provides an effective approach to leveraging private datasets for training more optimal models while maintaining data privacy. The combination of FL and LLMs presents a promising research direction for addressing data-related challenges in the future (Li et al., 2024). FATE-LLM (Fan et al., 2023) explored traditional tasks in federated learning fine-tuning for LLMs, while FwdLLM (Xu et al., 2023c) focused on improving memory and time efficiency during client training, thereby reducing the costs on client devices. FederatedScope-LLM (Kuang et al., 2024) emphasized federated instruction tuning using the FedAvg algorithm. OpenFedLLM (Ye et al., 2024) introduced a framework capable of completing both the SFT and RLHF training processes.

In our paper, we focus on SFT based on Non-IID datasets. To the best of our knowledge, we are the first to study this issue and propose a innovative framework that addresses both the client upload process and the global aggregation process.

## 3 Methodology

In this section, we will first introduce the problem that we face, and then propose our solution.

### 3.1 Preliminaries

LoRA (Hu et al., 2021) is employed to enhance the efficiency of fine-tuning by focusing on the internal rank variations that occur during parameter updates in the fine-tuning process. For fine-tuning a pre-trained model $\theta \in \mathbb{R}^{d \times k}$, LoRA keeps the pre-trained model's parameter matrix frozen and utilizes two lower-rank matrices, $\theta_{down} \in \mathbb{R}^{d \times r}$ and $\theta_{up} \in \mathbb{R}^{r \times d}$, to represent the update $\Delta\theta$. This process can be formulated as:

$$\theta' = \theta + \underbrace{(\theta_{down} \cdot \theta_{up})}_{\text{update}}$$

During the training process, $\theta$ remains frozen, while $\theta_{down}$ and $\theta_{up}$ are updated. Prior to training, we initialize $\theta_{up}$ using the Kaiming distribution and set $\theta_{down}$ to a zero matrix, ensuring that $\theta'$ initially equals $\theta$. Throughout training, the pre-trained model $\theta$ stays frozen, with updates applied only to $\theta_{up}$ and $\theta_{down}$. It is important to note that using $\theta_{down}$ and $\theta_{up}$ is not the only approach for decomposing $\Delta\theta$. Any low-dimensional decomposition method can be applied, as demonstrated in recent works such as Dettmers et al. (2023) and Rajabzadeh et al. (2024). In LLM training, LoRA is commonly used to reduce the computational cost associated with updating parameters. Typically, the base model $\theta$ is kept frozen, while only $\theta_{down}$ and $\theta_{up}$ of the $Q$, $K$, and $V$ matrices are updated.

In federated learning, each client $k$ trains its local model $\theta_k^i$ on its own dataset during the $i$-th iteration, and then sends the model to the server for global aggregation. After the aggregation, the client uses the updated global model $\theta^i$ to train and obtain the next local model $\theta_k^{i+1}$.

$$\theta \leftarrow \sum_{k=1}^{N} \frac{n_k}{n} \theta_k$$

where $\theta_k$ is the model of $k$-th client, $n_k$ is the dataset number of the $k$-th client, $n$ is the sum of the dataset number.

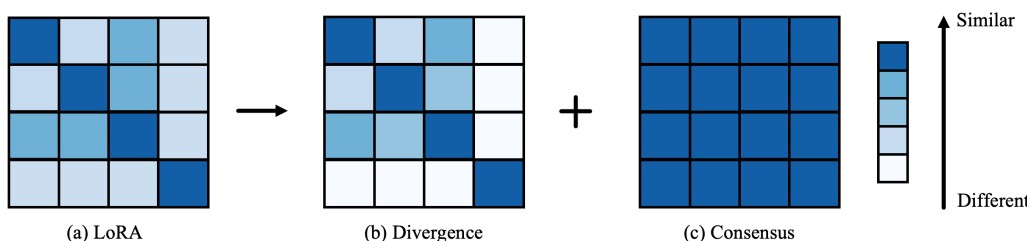

(a) LoRA          (b) Divergence          (c) Consensus

Figure 2: The similarity matrices between clients are shown as follows: (a) Similarity of LoRA matrices without decomposition, (b) Similarity of divergence matrices, and (c) Similarity of consensus matrices. The results indicate that *the differences between distributed LLM behavior become more pronounced after decomposition into consensus and divergence.*

In Fed LLM training, a common approach to reduce communication costs is to utilize LoRA, which updates only a subset of parameters instead of the entire model. In the FedAvg method with LoRA, the global model $\theta$ is updated as follows:

$$\Delta\theta \leftarrow \sum_{k=1}^{N} \frac{n_k}{n} \Delta\theta_k$$

where $\Delta\theta$ is the LoRA model of the global model, and the $\Delta\theta_k$ is the LoRA model of the k-th local client model.

We can compute the global model by adding the $\theta_{down} \cdot \theta_{up}$ to the frozen base model $\theta$, meaning that the global model $\theta'$ after training is given by:

$$\theta' = \theta + \Delta\theta = \theta + \theta_{down} \cdot \theta_{up}$$

where $\theta_{down} \cdot \theta_{up}$ represents the LoRA matrix corresponding to $\Delta\theta$.

However, two significant problems can arise with this training method: (1) The direct and indiscriminate aggregation of parameters from clients can lead to slow global convergence, particularly when working with Non-IID datasets. This issue may also result in knowledge bias, which can negatively impact the global model's generalization. (2) Some of the knowledge learned by local clients on their private datasets may not be relevant or important to the global model. These elements can potentially steer the global model toward a local optimum, hindering overall performance.

### 3.2 METHOD

#### 3.2.1 CONSENSUS-DIVERGENCE AGGREGATION

**Motivation.** In the current Fed LLM training process, the global model merely aggregates the LoRA matrices uploaded by the clients, overlooking both the interpretability of LoRA and the instability in global model aggregation that arises from client drift.

**Consensus-Divergence Decomposition.** Inspired by Weight Normalization (Salimans & Kingma, 2016), we decompose the client's uploaded parameters into magnitude and direction and update the direction vector using LoRA, as shown in the following formula, to investigate the distinct characteristics of a vector's mean in federated learning:

$$\theta^i = ||\theta^i||_c * \frac{\theta^i}{||\theta^i||_c}$$

where $||\theta^i||_c$ refers to vector-wise norm of $\theta^i$ across each column, and $\frac{\theta^i}{||\theta^i||_c}$ refers to the unit vector of $\theta_k^i$. The first part refers to consensus factor and the second part refers to divergence factor.

We train the clients on the Non-IID datasets, as described in Section 4, and aggregate them using the FedAvg method (McMahan et al., 2017). During this process, we calculate the similarity of the LoRA matrices across different clients and visualized it using a heat map, as shown in Figure 2.

Easily, we can find that the client's drift shows more clearly after splitting parameter matrices in weight and direction. In the splitting situation, we can find that the differences between the clients are displayed more prominently in Divergence Similarity Matrix and there is almost no difference between the clients in the Consensus Similarity Matrix.

Therefore, we can get two conclusion. (1) The decomposition method can better capture the differences between each client compared to base LoRA method. (2) We can get the real mean of the two matrices after decomposition: the direction matrix express **divergence characteristics** and the weight express the **consensus characteristics** of the model.

**Consensus-Divergence Aggregation.** In order to constraint the global aggregation, we introduce global similarity aggregation.

For the consensus vector $M^i = ||\theta^i||_c$ of the client in the $i$-th iteration, we calculate the average delta update of the client model. During the local training process, the local LLM extracts base knowledge from the local datasets. Averaging the differences in updates ensures that the new consensus factor absorbed by each client is more evenly integrated into the global model, following the format outlined below:

$$\Delta M^i = \Delta M^{i-1} + \frac{\sum_{k=0}^{N}(\Delta M_k^i - \Delta M^{i-1})}{N}$$

For the divergence matrix $V^i = \frac{\theta^i}{||\theta^i||_c} \in d \times k$, where $k < d$, can be decoupled by $V_{base}^i + V_{down}^i \cdot V_{up}^i$ and updates it using LoRA. At first, we expand the direction vector to obtain $d$ vectors of size $1 \times K$ from $V^i$. Then, for every single vector $v$, we calculate the similarity for different clients using cosine similarity follows:

$$\text{sim}(v_i, v_j) = \frac{v_i \cdot v_j}{||v_i|| ||v_j||}$$

where $v_i$ and $v_j$ refer to the same layer of the different clients.

For each client's vector $v_k$, we can calculate its average similarity $\bar{v}_k$. Since the cosine value reflects the degree of deviation of vectors from the global model in the divergence, we add a softmax activation layer before aggregation to determine the deviation weights of each vector within the overall divergence. Through this operation, we can obtain the weight proportion of each large language model's individual divergence within the overall consensus.

$$w_k = \frac{\exp\left(\frac{\bar{v}_k}{T}\right)}{\sum_{s=1}^{n} \exp\left(\frac{\bar{v}_s}{T}\right)}$$

Finally, we calculate the delta vector from the $(i-1)$ th to $i$ th iteration, getting the real global vector by adding weighted delta vector to the global divergence vector $V^{i-1}$ in the $i-1$ th iteration, with following the formula below. In this way, we construct a global divergence vector during the global update. On one hand, this approach ensures better compatibility with the consensus; on the other hand, capturing the divergence weights allows the global model to more accurately track the update directions driven by different clients, minimizing the negative impact of client knowledge drift. Consequently, the overall knowledge domain becomes more stable, enhancing the model's generalization ability.

$$\Delta V^i = \Delta V^{i-1} + \sum_{k=0}^{N} w_k \cdot (\Delta V_k^i - \Delta V^{i-1})$$

### 3.2.2 GAUSS-NOISE MASKING

**Motivation.** In existing methods, the local LLM uploads all parameters after each round of training. However, in real-world scenarios, not all LoRA parameters are fully updated; instead, updates primarily occur in certain key directions. Parameters that receive fewer updates can be considered less

important, which may slow down the aggregation of the global model and even negatively impact its generalization capabilities.

**Masking Generation And Upload** We calculate the importance of vector $k$ using the L2 norm, as shown in the formula below. A larger $\|D_k\|_2$ indicates that the client is in an active state during the current round of updates, while a smaller value suggests a more passive state during local updates. We consider more active vectors to contribute more energy to the local LLM, both in terms of consensus and divergence.

$$\|D_k\|_2 = \|L_k - G_k\|_2$$

where $L_k$ refers to the vector $k$ in the local model while the $G_k$ refers to the vector $k$ in the global model and $\|\cdot\|_2$ refers to L2 norm.

Then, we sort to select top $\alpha$ vectors $K_{update}$ for updating, while the rest of the vectors $K_{noise}$ will be filled with Gauss Noise. The whole parameters $U_k$ which will be uploaded follows the formula below.

$$U_k = \begin{cases} L_k, & \forall k \in K_{\text{update}} \\ L_k + \mathcal{N}(0, |D_k|), & \forall k \in K_{\text{noise}} \end{cases}$$

For each client, we upload parameters to the global after adding a Gauss-Noise mask layer. This method of updating through masking can effectively reduce the risk of local large language model client overfitting to localized knowledge.

The above explanation can be summarized by the detailed algorithmic process 1 below.

---

**Algorithm 1:** FedCDD

---

**Input:** Communication rounds $N$, participant scale $K$, $k^{th}$ client private model $\theta_k$, mask sparsity $\alpha$ and temperature $T$.

**Output:** The final global model $\theta^N$ at $N$ th round.

**for** $t = 1, 2, \cdots, N$ **do**

    *Client Side:* **for** $k = 1$ *to* $K$ *in parallel* **do**

        $f_k(\cdot) \leftarrow \text{ImportantSort}(\theta^{t-1}, \alpha)$ // Sort important vector for updating

        $f_k(\cdot) \leftarrow \text{GaussNoiseMask}(f_k(\cdot), \theta^{t-1})$ // Masking unimportant parameters

        $\theta_k \leftarrow f_k^t(\theta_k)$ // Calculate the new $\theta_k$ for uploading

    *Server Side:*

    $M_k^t, V_k^t \leftarrow \theta_k$ // Decoupling the $\theta_k$ into consensus and divergence

    // Consensus Aggregation

    $M^t = M^{t-1} + \frac{\sum_{k=0}^{N}(M_k^t - M^{t-1})}{N}$

    // Divergence Alignment

    $g(\cdot) \leftarrow \text{CaculateCosineSimilarity}(V^{t-1})$ // Build cosine similarity matrix

    $g(\cdot) \leftarrow \text{Softmax}(g(\cdot), T)$ // Calculate weight for vectors

    $V^t = V^{t-1} + g(V_0, V_1, ..., V_k)$

    $\theta^t \leftarrow M^t, V^t$

**return** $\theta^N$

---

## 4 EXPERIMENT

### 4.1 EXPERIMENTAL SETUP

**Datasets**. Following Ye et al. (2024), we train our model on the following datasets.

- Taori et al. (2023): The dataset used for fine-tuning the Alpaca model. Alpaca is a dataset of 52,000 instructions and demonstrations generated by OpenAI's text-davinci-003 engine.

| Methods | Generalization | | | | | Code | | Financial | | Math | | Average | |
|---|---|---|---|---|---|---|---|---|---|---|---|---|---|
| | MT-1 | MT-2 | Final Gen | Rank | Avg Rank | Score | Rank | Score | Rank | Score | Rank | Score | Rank |
| Base | 2.92 | 2.05 | 2.48 | 8 | 8 | 0.018 | 8 | 0.297 | 8 | 0.04 | 8 | 7.813 | 8 |
| FedAvg | 4.47 | 3.32 | _3.89_ | 2 | 2 | 0.048 | 7 | 0.345 | 5 | 0.05 | 7 | _12.120_ | 4.6 |
| FedProx | 4.29 | 3.30 | 3.79 | 4 | 4 | 0.079 | 5 | 0.284 | 7 | 0.13 | 3 | 11.874 | 4.6 |
| FedAvgM | 4.52 | 2.95 | 3.74 | 5 | 5 | 0.091 | 3 | **0.397** | 1 | 0.1 | 6 | 11.800 | 4 |
| Scaffold | _4.58_ | 3.10 | 3.84 | 3 | 3 | **0.100** | 1 | 0.297 | 6 | 0.12 | 5 | 12.036 | 3.6 |
| FedAdam | 4.45 | 2.91 | 3.67 | 6 | 6 | 0.085 | 4 | 0.381 | 3 | 0.122 | 4 | 11.627 | 4.6 |
| FedYogi | 4.46 | 2.90 | 3.67 | 7 | 7 | 0.061 | 6 | _0.382_ | 2 | **0.16** | 1 | 11.616 | 4.6 |
| Ours | **5** | 3.2 | **4.10** | 1 | 1 | _0.097_ | 2 | 0.351 | 4 | _0.14_ | 2 | **12.888** | 2 |

Table 1: Comparison of other methods on different evaluation methods. The best and second results are highlighted with bold and underline, respectively.

- Xiang Yue (2023): The dataset concerning the math field. Math instruct is compiled from 13 math rationale datasets, six of which are newly curated by this work. It uniquely focuses on the hybrid use of chain-of-thought (CoT) and program-of-thought (PoT) rationales, and ensures extensive coverage of diverse mathematical fields.

- CodeAlpaca-20k: The dataset concerning the code field. The 20K instruction-following data generated by the techniques Self-Instruct (Wang et al., 2022), with some modifications by author of the datasets.

- FinGPT: The specialized financial datasets used in FinGPT (Yang et al., 2023).

**Framework Setup**. We train our model based on the NousResearch's Llama-2-7b-hf within 200 rounds. And the we use different evaluation methods to test the performance of the model.

**Comparison Methods**. We compare FedCDD with several state-of-the-art methods in recently research and traditional FL: (1) **Base Model without SFT**. (2) **FedAvg**. (McMahan et al., 2017) the standard federated averaging algorithm, where updates from all clients are averaged at the server. (3) **FedProx**. (Li et al., 2018) An extension of FedAvg that introduces a proximal term to tackle heterogeneity across clients. (4) **Scaffold**. (Karimireddy et al., 2019) A control variate-based method designed to reduce the impact of client drift in federated learning with Non-IID data. (5) **FedAvgM** (Hsu et al., 2019) A momentum-based variant of FedAvg, which integrates server-side momentum into the federated learning process. (6) **FedAdam**. (Reddi et al., 2020) A federated version of the Adam optimizer. It adapts the learning rates at the server side using first and second-order moments of gradients, aiming to provide better performance in challenging federated settings. (7) **FedYogi**. (Reddi et al., 2020) An adaptive federated optimization method similar to FedAdam. Notably, recent studies on federated language model training are all based on the FL algorithm framework, where multiple LLM clients are run independently, and their parameters are simply aggregated. Therefore, in this test, we only compare the classic FL algorithms that have been widely used in peer research.

**Implement Details.** We provide the details from three views as:

- **Dataset Split**: We use datasets from four different domains as mentioned above, with each client randomly selecting 5000 labeled data points from its respective dataset for SFT.

- **Training Setting**: In the training process, we keep the learning rate $5e - 5$. In the first set, we set up the $\mu$ of FedProx 0.01. We repeat each experiment three times for each federated approaches to ensure the robustness and reliability of the results.

- **Evaluation Metric**: (1) **Generalization**: We use the first turn's score from MT-Bench (Zheng et al., 2023) as the primary evaluation metric to assess the general performance of different models (Ye et al., 2024). This score is the most critical in the overall evaluation. (2) **Contextual Understanding**: We use the final score from MT-Bench to evaluate the model's ability to understand context. MT-Bench comprises two turns of conversation, making it suitable for contextual testing. (3) **Code**: We use Human Eval (Chen et al., 2021) to evaluate the model's coding capabilities. (4) **Financial**: We utilize the MMLU dataset (Hendrycks et al., 2020) to evaluate the

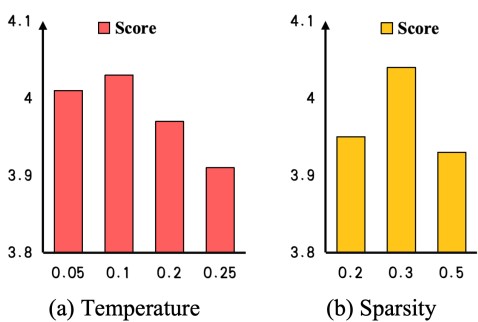

(a) Temperature    (b) Sparsity

Figure 3: **Hyper-parameter evaluation** on MT-Bench, focusing on (a) the temperature of Consensus-Divergence Aggregation and (b) the sparsity of Gauss-Noise Masking. Higher scores indicate better performance.

| Client | Global | Generalization | | |
|--------|--------|------|------|-------|
|        |        | MT-1 | MT-2 | Final |
| ✗ | ✗ | 4.47 | 3.32 | 3.89 |
| ✓ | ✗ | 4.70 | 3.39 | 4.04 |
| ✗ | ✓ | 4.70 | 3.41 | 4.06 |
| ✓ | ✓ | **5.00** | 3.20 | **4.10** |

Table 2: **Ablation study** of key components in MT-Bench. The score of MT-1 indicates the general ability of the large language model and the final score indicates the contextual understanding.

model's financial knowledge, specifically selecting the finance-related domains for this assessment. (5) **Math**: The GSM8k dataset (Cobbe et al., 2021) is used to test the model's mathematical abilities. For each evaluation, we either use GPT-4o to assess the model's responses in an open-ended environment or compare them against standard answers according to the requirement of the benchmark. To account for the variability in large model outputs, each experiment is repeated three times to ensure robustness and reliability. After the above evaluation, we use average rank to display the comprehensive capabilities of the model.

## 4.2 EXPERIMENT RESULTS

**Performance Comparison.** Table 1 presents the performance of different methods in traditional FL compared to our FedCDD approach. The results demonstrate that FedCDD outperforms the other methods, highlighting its effectiveness in large language model training within a federated learning context. Traditional methods like FedAvg and FedProx fail to effectively aggregate client consensus and align divergences, resulting in a degradation of model performance. In contrast, FedCDD successfully preserves the model's generalization capabilities under these conditions. Specifically, our framework demonstrates a significant improvement in generalization and consistently maintains an advantage across various specialized domains.

## 4.3 DIAGNOSTIC EXPERIMENTS

**Key Components.** We conduct an ablation study on the key components of our method using the MT-Bench on the four diverse datasets with the optimal hyper-parameters of the different key components. The results, demonstrating the effectiveness of each component, are presented in Table 2. Both components can enhance the performance of the global model and achieve optimal results when combined.

**Hyper-Parameters.** We conduct a hyper-parameter ablation analysis using MT-Bench, focusing on the general performance of the model, as shown in Figure 3. The analysis examines two key hyper-parameters: the masking sparsity $\alpha$ and the temperature $T$ in the global aggregation process. We observe that variations in temperature within a narrow range do not significantly affect the average results, likely because optimal performance can be achieved within an appropriate range of $T$. However, masking sparsity $\alpha$ has a much larger impact, as deviations in $\alpha$ led to performance fluctuations. Specifically, if $\alpha$ is too large, the global model may lose important updates from clients, while setting $\alpha$ too low results in an overemphasis on less significant client information. In most of our experiments, we set the default values to $T = 0.1$ and $\alpha = 30\%$.

## 5 DISCUSSION AND LIMITATION

(1) Our method is based on LoRA, which does not enhance model performance as effectively as full parameter fine-tuning. However, training with LoRA significantly reduces training time and is easily adaptable to various downstream tasks. In the future, we aim to develop a method that combines the peak performance of full fine-tuning with the flexibility and cost-efficiency of LoRA.

(2) Our method is rooted in the field of federated large language model training, but its underlying principles are broadly applicable. In the future, we will continue to explore this approach to extend its applicability to more specific domains within federated learning.

## 6 CONCLUSION

In this paper, we are pioneers in innovatively exploring the large language model fine-tuning in the federated learning on the Non-IID datasets. Additionally, we are the first to establish a new algorithm for federated LLM training. We propose a novel framework called FedCDD, an effective federated consensus-divergence decoupling for LLM fine-tuning. We decouple the updates of LoRA into divergence and consensus, seizing the subtle updates of the LLM updates. At server level, we align the divergence using cosine similarity and aggregate consensus of LLM, enhancing the generalization ability of the global model. At the client level, we apply Gauss-Noise Mask to the parameters being updated, avoiding the client' local knowledge affecting the generalization ability of the global model due to the over-fitting. This method has demonstrated effectiveness and robustness across multiple experiments. We hope this work offers a novel perspective for future research on federated large language model fine-tuning.

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
