# OpenReview forum: "Divide And Conquer: Efficiently Decoupling Consensus And Divergence For Federated Large Language Model Fine-Tuning"
_ICLR.cc/2025/Conference — Submitted to ICLR 2025_

### Official Review · Reviewer_SiCz · 2024-10-23

**Soundness:** 2
**Presentation:** 1
**Contribution:** 2
**Rating:** 3
**Confidence:** 4

**Summary:**

This paper addresses the non-iid problem of federated LLM LoRA fine-tuning by decomposing the low-rank matrix into consensus and divergence components. Each factor is updated differently to improve the generalization of federated learning. The approach shows improved generalization on some datasets. However, the paper lacks clarity in several areas, including the motivation for key design choices, insufficient comparison with related work, and the limited experimental results.

**Strengths:**

S1. This study focuses on an important problem - non-iid federated fine-tuning - which is prevalent in practical scenarios.

**Weaknesses:**

W1. The authors claim that this is the first study on non-iid federated LLM fine-tuning, which appears to be an overstatement. Related work [1] is not discussed or compared. Furthermore, most federated LLM fine-tuning studies have performed non-iid experiments (with different Dirichlet parameter $α$ values), even without specific non-iid designs.

W2. While evaluating divergence by comparing directions is straightforward, it is not clear why $||θ^i||_c$ is referred to as consensus and why consensus can be directly averaged.

W3. The rationale for using average cosine similarity as a weight in model averaging is not clearly explained. Suppose two clients, A and B, have identical data and model parameters, and another client, C, has low similarity to A or B. This implies that updates from clients A and B will dominate the training process due to the high similarity. It is unclear why this outcome is expected, and further justification is needed to clarify the reasoning behind this approach.

W4. The experimental results do not convincingly demonstrate the superiority of the proposed method. The proposed method performs best only on MT-1 (it is unclear if MT-final is an independent evaluation set).

W5. The experimental scope is limited, with inadequate ablation studies. It is unclear what "client" and "server" represent in Table 2. The contribution of consensus and divergence to the performance improvement is not demonstrated, and the effect of imbalance level (Dirichlet parameter) is not studied.

**Minor Suggestions**

M1. Lines 107, 133: "LLama" should be "LlaMA".

M2. The dataset names in Section 4.1 do not match the names in Table 1, making it unclear what each column refers to.

**References**

[1] Cho, Yae Jee, et al. "Heterogeneous LoRA for federated fine-tuning of on-device foundation models." International Workshop on Federated Learning in the Age of Foundation Models in Conjunction with NeurIPS 2023. 2023.

**Questions:**

Q1. Why is $θ_i$ calculated by columns instead of rows of the parameter matrix? Does this apply to both $θ_{up}$ and $θ_{down}$, which have different shapes?

Q2. How is the average of $v_k$ calculated? Is it an average over clients or over layers?

Q3. Is the Gaussian mask updated at each step/epoch?

Q4. What is ImportantSort sorted by? How is importance defined?

Q5. Is this method specifically designed for LoRA? Since it operates on the parameter matrix, can it be adapted for general federated learning?

Q6. How does the proposed method compare to traditional non-iid federated learning approaches, e.g. SCAFFOLD, FedProx?

---

### Official Review · Reviewer_CHcf · 2024-10-26

**Soundness:** 2
**Presentation:** 2
**Contribution:** 2
**Rating:** 3
**Confidence:** 3

**Summary:**

This paper proposes a federated learning algorithm that considers both the consensus and divergence of the local training and global aggregation. The algorithm employs some decomposing computations to decouple the two factors in updating parameters and the Gaussian mask to regularize the local model uploads. Some experimental results are provided to demonstrate the effectiveness over the existing algorithm.

**Strengths:**

1. This paper demonstrates the effectiveness of decomposing the local client uploads into consensus and divergence parts.
2. Some preliminary experiment results can demonstrate that the proposed method can finetune an LLM to a better state (in terms of average ranking) than the existing FL algorithms.

**Weaknesses:**

1. The notations and the relation between the notations in the paper need to be further revised and be more concise. The notations in the current version make it hard for the reader to understand the core algorithm in the paper. Section 3.2 contains many notations in addition to the $\theta$ used for LoRA, including $M$, $N$, $v$, $V$, $L_k$ and $G_k$. It is hard to understand how those notations relate to the parameter of LoRA. Although some explanation efforts can be observed in the subsection, it brings more questions, such as what the "vector" refers to or corresponds to in LoRA.
2. Some key parts of the algorithm may need to be further illustrated, including what are the parameters that local clients share and which part of the computation is done on the client/server, respectively.
3. The motivation for the Gauss-noise masking is unclear and somehow not self-contained. In the "motivation" paragraph, it mentions the parameters with fewer updates can slow down the global convergence; however, introducing noise on those parameters can not solve such a problem in common belief but can even make the convergence slower. (This can refer to differential privacy papers where Gaussian noise is usually applied.) More theoretical or empirical support or existing literature references are required to justify such a design.
4. The experiment setting and explanation is not persuasive. 1) It is unclear why the scores from different datasets can be added together, given that their scales already seem different. 2) Only one set of experiment datasets are evaluated. Although such data setting can be one of the most heterogeneous settings, it may expected to show the effectiveness of the algorithm on more datasets, such as different clients have different coding languages. 3) It is unclear what "client" and "global" mean in Table 2. 4) Only the evaluation on MT-bench is presented, but the results on Code and others are not presented.

**Questions:**

Please response the points in Weaknesses.

---

### Official Review · Reviewer_NHm3 · 2024-11-01

**Soundness:** 2
**Presentation:** 2
**Contribution:** 1
**Rating:** 1
**Confidence:** 5

**Summary:**

The authors propose a framework aimed at conducting LLM fine-tuning in federated environments. They argue that LLM fine-tuning model parameters cannot be simply aggregated due to client heterogeneity.

**Strengths:**

This is a new and exciting field with sound motivations, and hence, a lot of room to maneuver. The literature referred to are also pretty new.

**Weaknesses:**

1. Many claims sound sensible on paper, but are not supported either empirically or theoretically. An example is how current Fed LLM training overlooks "interpretability of LoRA and the instability" that arises from client drift. We don't see this demonstrated rigorously in the paper. For example, how does the extent of client drift affect the results?

2. The idea of consensus-divergence decomposition is just another name for well-established methods that look at the parameter update's magnitude and direction. It just feels like the authors now apply it to LoRA. As to how this method is made to be specific to LoRA, nothing much was  suggested.

3. The paper looks incomplete. The experiments are few, for a paper that has no theoretical backings.

**Questions:**

1. Provide convergence guarantees to show that the method mitigates instability and improves interpretability (as the authors claim)

2. Improve the experiment sections to include: different models, data distributions (extents of heterogeneity), ablation study of number of clients etc.

---

### Official Review · Reviewer_nvfV · 2024-11-07

**Soundness:** 2
**Presentation:** 3
**Contribution:** 2
**Rating:** 5
**Confidence:** 2

**Summary:**

The paper introduces a novel framework for federated fine-tuning of large language models (LLMs) called Federated Consensus-Divergence Decoupling (FedCDD). This framework aims to improve the performance of LLMs in heterogeneous environments by decoupling client updates into "consensus" and "divergence" components. It incorporates a Gaussian-Noise Masking technique to selectively update significant parameters and minimize overfitting to domain-specific features.

**Strengths:**

The FedCDD framework improves the generalization ability of the model in different datasets by effectively managing the consistency and difference in parameter updates. Gaussian noise masking reduces computational cost and communication overhead by selectively updating important parameters, which also helps maintain privacy to a certain extent because only necessary information is transmitted on the network.

**Weaknesses:**

Insufficient methodological innovativeness: the inclusion of Gaussian-Noise Mask by simply judging the importance of the parameters through L2 Norm seems to be simplistic, and the authors do not explain clearly why this is effective. Even if it is effective, the Gaussian-Noise Mask only masks the non-important information, so will the important information be easily recognized. Does this really protect the privacy of important information?

Experimental limitations: The experimental setup of this paper is not robust enough to convincingly demonstrate the effectiveness of the proposed method. It lacks ablation studies, which are crucial to understand the contribution of individual components such as consensus-divergence aggregation and Gaussian noise masking. In addition, the benchmark comparison is outdated and mainly involves studies before 2020, which do not reflect the state-of-the-art in federated learning and LLM fine-tuning.

Unexplored trade-offs: In federated learning, there are inherent trade-offs between model accuracy, data privacy, and fairness among participating clients. These trade-offs are not discussed or analyzed in this paper.

**Questions:**

Refers to the weakness.

---

### Meta-Review · Area_Chair_xi23 · 2024-12-14

**Metareview:**

## Summary of Contributions

This paper studies Federating fine-tuning of LLMs. One way to do this is for each client to compute their update, via e.g. LoRA, and aggregate these updates. The paper proposes a different way to aggregate the updates, based on "consensus-divergence composition", and a Gaussian noising mechanism to help with overfitting. Experiments are conducted comparing the proposed methods with known methods.


## Strengths
- Federated fine-tuning of LLMs with non-i.i.d. data is an important and well-motivated setting.

## Weaknesses
- The paper fails to explain intuition or justification for most core concepts in the paper. For example, it is not clear why it makes sense to aggregate the "consensus" and "divergence" part separately. Similarly, it is not clear what Gaussian-noise masking actually achieves, given that no noise is added to the most important updates, and how it is different from adding noise to all the updates.

- The experiments are not comprehensive. And even in ones that are included do not seem to clearly show the superiority of the proposed methods, as other methods also achieve better results for certain metrics (in Table 1).

- There is no theoretical backings of the methods.

## Recommendation

Given the weaknesses, we believe that the paper is not ready to be accepted in the current state, and we recommend rejection.

**Additional Comments On Reviewer Discussion:**

The authors didn't reply at all during rebuttal, so there was no additional discussion.

---

### Decision · Program_Chairs · 2025-01-22

Reject